# Heat Accumulation-Induced Surface Structures at High Degrees of Laser Pulse Overlap on Ti6Al4V Surfaces by Femtosecond Laser Texturing

**DOI:** 10.3390/ma16062498

**Published:** 2023-03-21

**Authors:** Farkhod Babadjanov, Uwe Specht, Thomas Lukasczyk, Bernd Mayer

**Affiliations:** 1Fraunhofer Institute for Manufacturing Technology and Advanced Materials IFAM, Wiener Str. 12, 28359 Bremen, Germany; 2Faculty of Production Engineering, University of Bremen, 28359 Bremen, Germany

**Keywords:** ultrashort pulsed laser, laser pulse overlap, Ti6Al4V, structures for drug delivery

## Abstract

In this work, femtosecond laser pulses at high repetition rates were used to fabricate unique microstructures on the surface of Ti6Al4V. We investigated the influence of pulse overlap and laser repetition rates on structure formation. Laser texturing with a high degree of overlap resulted in melting of the material, leading to the formation of specific microstructures that can be used as cavities for drug delivery. The reason for melt formation is attributed to local heat accumulation at high repetition rates. Such structures can be fabricated on materials with low thermal conductivity, which prevent heat dissipation into the bulk of the material. The heat accumulation effect has also been demonstrated on steel, which also has low thermal conductivity.

## 1. Introduction

One of the biggest challenges in manufacturing implants is to create a bio-interactive surface that bacteria cannot attach to, while at the same time producing an implant with strong mechanical properties. Titanium and titanium alloys are known for their high corrosion resistance, biocompatibility, low modulus of elasticity, and strength [1]. Due to these features they are widely used metallic materials in the field of orthopedics and implant dentistry [2,3,4]. The topography of implants play a key role for cellular response and, consequently, the osseointegration process [5]. Enhanced and faster osseointegration was observed on nano-scale-modified surfaces [6]. In the past, several studies have been undertaken to modify the surface of titanium, such as sandblasting [7], chemical etching [8], and different coating techniques [9]. The conventional methods of surface modification, such as plasma electrolytic oxidation, thermal oxidation, and corundum grid blasting, can severely reduce the fatigue performance of implants [10,11]. Compared to conventional processes, the laser is energy-saving, contactless, wear-free, and generates no environmentally harmful waste. Recontamination of the surfaces is also avoided, and the structures produced can also be applied locally with micrometer precision [12]. Laser surface texturing has proven to be a promising method to create various micro- and nanometer-scale structures with reduced surface contamination, while being environmentally friendly [13,14,15]. These structures enable fabrication of functional materials in the field of electronics, mechanics, wettability, and optical response [16,17,18]. Studies have shown that laser texturing enables fabrication of undulating grooves [19], bumps/spikes [20], and holes [21]. In addition, texturing with femtosecond lasers results in a smaller heat-affected zone (HAZ) compared to nano- or picosecond lasers [22], which affects the mechanical properties of the material by changing the microstructure of the material and introducing residual stresses.

In recent years, laser texturing has become a common method of surface modification in medical technology [23,24,25]. Cunha et al. showed reduced bacterial adhesion on laser-induced periodic surface structures on titanium surfaces [26]. Dumas et al. demonstrated that micro-pits and nano-ripples generated by a femtosecond laser enhance the spreading speed of mesenchymal stem cells and improve osteogenic potential [27]. Shan et al. observed that laser-induced periodic surface structures (LIPSS) and nano-pillars can be applied for increasing the adsorption of proteins, which are responsible for bone growth and matrix mineralization [24,25].

After implantation, the patient’s immune system is weak, which allows bacteria to spread rapidly. Hence, the suppression of a bacterial attack is important during the first hours and days after implantation. Pathogen microbes causing infections on the implant area are delivered through the bloodstream, for example, from inflammation of a tooth leading to the septic implant loosening [28]. For this reason, researchers are focused on the development of strategies with antibiotics with localized and controlled drug release kinetics [29]. Antibiotic or drug-loaded fibers are broadly used materials for the elimination of bacterial adhesion [30]. In most cases these pathogens are Staphylococci, especially Staphylococcus aureus [31,32]. Recently, a new approach for drug delivery was demonstrated on implants textured with a nanosecond laser, where the laser-induced cavities were filled with a drug for a burst release effect [23]. It was shown on animals and humans that local implementation of antibiotics has a higher anti-inflammatory potential than systematic application of antibiotics because of a higher concentration [33,34,35]. A good control of drug release properties was achievable via microstructures with undercuts. Therefore, the combination of modified surface topography together with a drug-loaded system is believed to have a great potential in the field of implantation.

In this paper, we show that structures fabricated at a high degree of fs-laser pulse overlap and high repetition rates results in a unique form of structures. Ti alloy implants that have micro-cavities on their surface can be used in the implant industry. Micro-cavities can be filled with drugs before implantation, so that the drugs act locally after implantation and prevent the adhesion of bacteria. The formation of such structures is a result of a complex material interaction with an incident laser beam that is dependent on electron–phonon (e–ph) coupling constant, thermal conductivity, and other physical properties of the material, wavelength, and polarization of the incident light, machining environment, and laser parameters, such as the laser fluence, laser absorptivity, and degree of pulse overlap [36]. Several studies have investigated the influence of these features on structure formation. Experiments were performed using single spot ablation as well as multiple spots at a different degree of pulse overlap and laser fluence [36,37,38,39,40]. Yet again, most of these studies focus on structures fabricated at a pulse overlap up to 90%, as laser texturing with a high degree of pulse overlap is not industrially lucrative. However, the rapid development of ultrafast lasers in recent years enables laser texturing with a high degree of overlap. Moreover, the influence of high repetition rates have not been examined thoroughly in previous studies, which also strongly affect the formation of structures on Ti6Al4V. The results of our extensive investigations indicate that the fabrication of such structures is due to local heat accumulation at high repetition rates, enabling also the generation of microstructures with undercuts. In order to prove it, Cu and steel were also laser textured to confirm our assumptions.

## 2. Materials and Methods

### 2.1. Materials

The investigations were performed on Ti6Al4V samples with a dimension of 2 cm × 2 cm × 1 cm (L × W × H) with an average surface roughness Sa of approximately 0.95 ± 0.03 µm (according to DIN EN ISO 4287). Ti6Al4V plates were purchased from ARA T Advance GmbH (Dinslaken, Germany) and correspond to the requirements of ASTM F136. In the second row of experiments, pure copper (Bikar Metalle, Bad Berleburg, Germany) and stainless steel (Edelstahl- & Metallhandelsgesellschaft mbH, Stuhr-Brinkum, Germany) samples were also laser textured. Prior to laser texturing, all samples were cleaned in an ultrasonic bath (Elmasonic S30H, Elma Schmidbauer GmbH, Singen, Germany) in acetone for 5 min. Drying was applied with dust-free wipes and compressed air.

### 2.2. Laser Texturing

Laser texturing of samples was performed by a Yb-doped femtosecond fiber laser YLPF-10-500-10-R (IPG Photonics Corporation, Oxford, MA, USA). The laser system delivers a beam at a wavelength of 1030 nm with pulses in the range of 120–650 fs and operates at a pulse repetition rate between 150 kHz and 1 MHz. Spatial intensity distribution of the beam is nearly equal to a spatial Gaussian profile (beam parameter product M^2^ ~ 1.4) with a maximum pulse energy of 20 µJ. The laser beam is focused on the surface of the target with a two-axis scan head-type Fiber Rhino (Novanta Europe GmbH, Wackersdorf, Germany) equipped with a F-theta lens with a focal length of 160 mm, leading to a circular focus diameter of *d_f_* = 30 µm. The laser texturing parameters were varied over a wide range to yield various microstructures and to investigate their effect.

The laser pulse overlap (*PO*) was calculated from Equation (1):(1)$$PO\;=(1-\frac{v_{s}}{d_{f}\times f_{\mathrm{rep}}})\;\times \;100\%$$
where $v_{s}$ is the scanning velocity, $d_{f}$ the circular focus diameter, and $f_{\mathrm{rep}}$ the repetition rate. The line spacing was kept constant and equal to 70 µm. The micromachining was performed at $f_{\mathrm{rep}}$ = 150 kHz, 500 kHz, and 1 MHz. Variation of *PO*s were achieved by an adjustment of the scanning velocity according to Equation (1). The used processing parameters are listed in Table 1.

After analyzing the results of the structures fabricated with the previously listed parameters, we decided to perform a second row of experiments varying the pulse energy at a constant pulse overlap and pulse repetition frequency for the treatment of steel and copper. These laser parameters are listed in Table 2.

### 2.3. Surface Characterization

The resulting micro- and nanostructures were analyzed by scanning electron microscopy in SE-mode at an acceleration voltage of 10 kV. (Phenom XL Thermo Fisher Scientific, Waltham, MA, USA). The profile of the resulting structures was measured with a confocal laser scanning microscope (Keyence VK9700, Osaka, Japan).

## 3. Results and Discussion

### 3.1. Microstructure Formation on Titanium at High Pulse Repetition Rates

Figure 1 shows an overview of the surface structures generated via the laser parameters listed in Table 1. Images were acquired via SEM measurements. All experiments led to the formation of surface structures with different morphology depending on the applied laser parameters.

When comparing the structures obtained at repetition rates of 150 kHz, 500 kHz, and 1 MHz, note that the only difference between the applied laser parameters is the time interval between successive laser pulses, which results from the applied repetition rate. The time interval between pulses at repetition rates of 150 kHz, 500 kHz, and 1 MHz accounts to 6.6 µs, 2 µs, and 1 µs, respectively. These time intervals are an important factor that allow us to demonstrate the effect of heat accumulation as a function of the applied repetition rate. In turn, the laser fluence was kept constant at 1.4 J/cm^2^ during these experiments on Ti alloy.

From the comparison of the SEM images, it can be concluded that the applied repetition rate had a great influence on the structures formed, although the applied laser fluence and number of pulses were kept constant. For example, the structures fabricated at *PO* = 98.80 and *PO* = 99.10 with different repetition rates show that a slight change in *PO* can result in a completely different type of structure. The textured surfaces at *PO* = 99.80 with repetition rates of 150 kHz and 500 kHz are relatively similar, although the surfaces textured with a repetition rate of 500 kHz have small particles that resemble redeposition particles. The surfaces textured with the same *PO* at 1 MHz again differ slightly from those textured at 150 kHz and 500 kHz. A SEM image shows that the textured lines have micro-holes along a grid line, and the textured area looks smoother compared to its original surface. When comparing *PO* = 98.80% and *PO* = 99.10%, no changes are observed on surfaces textured at a repetition rate of 150 kHz. Surfaces textured with a repetition rate of 500 kHz and 1 MHz, by contrast, differ significantly. Surfaces textured at *f* = 500 kHz yielded solidified bump-like structures with periodic micro-holes, while the surface textured at *f* = 1 MHz resembles shallow trenches with micro-holes that were not as frequent as those formed at *f* = 500 kHz. A further increase in *PO* to 99.50 at *f* = 150 kHz showed increased material removal leading to the formation of trenches on the Ti alloy surface, and the surface fell below the original surface level as shown in Figure 1. Laser texturing of the surface at *f* = 500 kHz resulted in enlargement and coarsening of the structures, while the surface textured at *f* = 1 MHz led to the formation of deeper trenches. Comparison of representative SEM images of structures fabricated at *PO* = 99.10 and *PO* = 99.50 at *f* = 500 kHz shows that increasing *PO* leads to increased formation of molten material with subsequent resolidification. When comparing surfaces textured at *PO* = 99.10 and *PO* = 99.50 at *f* = 1 MHz, it can be seen that increased *PO* leads to increased material removal as the number of laser pulses per area increases with increasing *PO*. When comparing trenches made at *PO* = 99.50 with different repetition rates, the edges of the trenches made at *f* = 1 MHz look smooth and resolidified, while those made at *f* = 150 kHz are sharp-edged, indicating clear material removal. Increasing *PO* to 99.58 at *f* = 500 kHz showed no difference between structures made with *PO* = 99.50, while trenches fabricated with *f* = 150 kHz became deeper. In turn, the trenches at *f* = 1 MHz became deeper, and a more molten material solidified at their edges. SEM images of structures made with *PO* = 99.66 and *PO* = 99.75 show that the trenches became even deeper at *f* = 150 kHz, while at *f* = 500 kHz no visible change in the structures is observed. In the case of *f* = 1 MHz at *PO* = 99.66, no resolidified material was observed at the edges of the tranches, while at *PO* = 99.75 the trenches became deeper, and the amount of resolidified material at the edges increased. It appears that at high *PO* levels and high repetition rates, the formation of excess molten material leads to the formation of unique structures, and the molten material is driven out of the inside of the trenches during laser texturing. Even if the applied fluence is the same, the time interval between pulses seems to play an important role in the formation of such structures. To the best of our knowledge, such structures are reported for the first time. A further increase in *PO* resulted in the disappearance of the trenches when repetition rates of 150 kHz and 500 kHz were used. The reason for this could be a slow laser scan speed that results in a dense plasma plume, which does not allow the subsequent laser pulses to reach the surface of the sample. At *f* = 1 MHz and *PO*s up to 99.86%, a larger amount of resolidified material accumulated at the edges of the trenches, and at *PO* = 99.90 and above, the trenches disappeared completely. It appears that at high repetition rates, the textured surface melts and resolidifies at the textured areas of the sample. Overall, the size and morphology of the structures depends not only on the fluence but also on the laser repetition rate. The main structure types to be generated are micro-holes and bump-like structures. It can be clearly observed that at higher repetition rates melt formation is more pronounced.

To summarize the evolution of surface topography during laser texturing, we performed confocal laser scanning microscopy (CLSM) measurements on textured samples. The results of the CLSM are shown in Figure 2.

Figure 2a shows an example of the characterization process of CLSM images. The exemplary surface visible in the image was fabricated at *f* = 1 MHz and *PO* = 99.90, with the black line representing the line scan profile at a representative position perpendicular to the fabricated laser trenches. The corresponding line scan profile is depicted below the CLSM image. Line scan measurements were conducted five times on each resulting structure at extreme values, i.e., at the lowest positions in the case of pure trenches, where no melt formation is observed, and at the highest positions, where the structures are a result of melt formation and resolidification. The local extreme values were measured in each line profile and are averaged for each *PO* value of the laser fabrication process. Figure 2b summarizes the evolution of the surface topography during laser texturing with increasing *PO* at a repetition rate of *f* = 150 kHz, *f* = 500 kHz, and *f* = 1 MHz. The x-axis indicates the degree of *PO*, while the y-axis provides information on the averaged extreme values relative to the original sample surface at a given *PO*. The red line in Figure 2b is the original surface level of the sample, i.e., the level of the non-textured sides of the sample, and is equal to 0.

As can be seen from the graph, the transition from pure trenches to melt-induced structures occurs earlier at a repetition rate of *f* = 500 kHz. At *f* = 150 kHz or *f* = 1 MHz, by contrast, the structured trenches become even deeper. The later transition of the trenches into the melt at *f* = 1 MHz could be due to the faster scan speed compared to *f* = 500 kHz, where heat dissipation into the bulk could play an important role in the formation of structures. Nevertheless, the transition to melt-induced structures also occurs at *f* = 1 MHz, and the height of the resulting structures is even larger than those induced at *f* = 500 kHz. This could indicate that the amount of melted material is larger at *f* = 1 MHz than at *f* = 500 kHz. Such behavior could be caused by shorter time intervals between laser pulses at *f* = 1 MHz, which could lead to local heat accumulation. When the *PO* was further increased, at *f* = 150 kHz, the trenches became even deeper and did not result in excessive melting of the material leading to the formation of preferred structures, as was the case at *f* = 500 kHz and *f* = 1 MHz. Nevertheless, a slight transition to melt-induced structures can be observed at *PO* = 99.86 at *f* = 150 kHz. However, a look at the SEM image at *PO* = 99.86 (see Figure 1) shows that these structures are not comparable to those induced at *f* = 500 kHz and *f* = 1 MHz. In addition, a further increase in *PO* to 99.90% at *f* = 1 MHz shows that the structures begin to collapse. The reason for this behavior could be the increasing amount of molten material with increasing *PO*, which does not allow the molten mass to move from the textured areas to their edges, as is the case with *PO* = 99.80. It should be noted that at such high *PO*s, the role of plasma shielding also becomes significant, which we discuss further. Overall, with the help of this graph, we can choose a texturing regime depending on a repetition rate that allows us to produce desired structures.

When discussing the generation of surface structures, it is important to distinguish between single and multiple laser pulse texturing. In the case of a single pulse texturing, the plasma plume is formed within ten picoseconds, and a combination of atoms, ions, and particle agglomerates are ejected [41]. In multiple laser pulse texturing, the subsequent pulses interact with the plasma cloud generated by the previous laser pulse, resulting in inefficient energy deposition known as plasma shielding [41,42]. A high-density plasma cloud was observed even after a few nanoseconds at fluences above the ablation threshold [43]. In addition, the interaction of the subsequent laser pulses with the plasma leads to redeposition of the ablated material on the material surface [44]. Since we use multiple pulses in our experiments, plasma shielding and heat accumulation have an influence on the formation of structures.

Moreover, two ablation regimes can be distinguished for femtosecond laser ablation, gentle and strong [40,45]. The gentle ablation regime is defined by the optical penetration depth, while strong ablation governed by the effective thermal penetration depth. Strong ablation is accompanied by the formation of a high amount of molten material, compared to gentle ablation, and is caused by phase explosion [39,40,46]. Considering these ablation mechanisms, we can conclude that the structures formed at *f* = 150 kHz are the result of gentle ablation, while at *f* = 500 kHz and *f* = 1 MHz, the ablation is predominantly strong. Another important aspect is the laser beam profile. In our experiments, we used Gaussian-shaped intensity distribution, which results in strong ablation regime in the center of the spot whereas the intensity of the laser beam in the surrounding area is much lower. This leads to a preferential displacement of the melt layer from the center of the textured area to its peripheral areas, where the melt temperature is lower. It should be noted that a different meld displacement behavior is expected for laser beams with different energy distributions, for example, flat-top-type systems.

Previously, it was also shown that multiple pulse femtosecond ablation leads to local heat accumulation, which causes the material to melt [37,47]. For instance, melt layer induced by a single femtosecond laser on copper persists for up to 200 ps before it resolidifies [48], while on Ti alloy, it persists up to 300 µs [36]. During machining of Cu with overlapping pulses, the pulse intervals at both 150 kHz and 500 kHz are considerably long, and consequent pulses impact the resolidified surface. The resolidified surface contains also a lot of structural defects, which can enhance the absorption of consequent pulses [39]. Liu et al. investigated the generation process of micro-holes on the surface of aluminum by femtosecond laser texturing [49]. They attributed this phenomenon to laser-induced melt formation and the Marangoni effect, in which the melt is driven from the center of the molten pool to the edge, where it solidifies. Sedao et al. investigated the influence of ultrafast laser processing of Ti alloy and steel [50]. They report that the thermal energy is stronger confined on the surface of Ti alloy and steel due to higher heat penetration depths and a larger absorbance coefficient compared to Cu. Another reason for the formation of melt is strong e–ph coupling constant of Ti alloy (see Table 3), which allows a fast energy transport of laser pulses into the material, but heat dissipation is insufficient due to low thermal conductivity. As a result, local heat accumulation and melting of the material is favored, evoking course microstructure formation. Moreover, numerical models also demonstrated that excess energy of laser pulses remain in the surface of ablated material and can lead to local heat accumulation at high repetition rates [51].

### 3.2. Influence of Substrate Material on Melt Formation

To demonstrate the influence of the e–ph coupling constant and thermal conductivity of metals on melt formation, we textured steel and Cu. The laser parameters are listed in Table 2. The samples were textured with an increasing laser fluence to show that the ablation threshold was exceeded, confirming sufficient energy deposition to ablate the material. The SEM images show that melt formation is observed when steel is textured, as was the case with the Ti alloy (see Figure 3).

Cu texturing, by contrast, shows the absence of melting and resolidification of the material, which confirms our theory of local heat accumulation. Even when the fluence was increased to 2.42 J/cm², no molten material was observed, but the trenches became even deeper. It proves that materials with high thermal conductivity tend to melt less at high repetition rates because heat can be dissipated into the bulk more quickly, while materials with low thermal conductivity begin to melt due to local heat accumulation.

## 4. Conclusions

This paper presents a comparative study on laser surface texturing of Ti6Al4V at high repetition rates and a high degree of pulse overlap (*PO*) using ultrashort laser pulses. With an increasing *PO* and a repetition rate, various surface structures were obtained. The reason for the formation of such structures is attributed to local heat accumulation at high repetition rates, which leads to melting of the material during laser texturing.

In the case of Ti alloy and steel, the reason for the formation of melt is their strong e–ph coupling constant, which allows a fast energy transport of laser pulses into the material lattice, but heat dissipation is insufficient due to low thermal conductivity, by contrast, has a high thermal conductivity and is less prone to melting at high repetition rates because the heat dissipates faster into the surrounding bulk. These effects have to be taken into account for specific laser system and material combinations, especially for novel developments in high-repetition ultra-short pulse lasers. Limited investigations were able to show such phenomena [52]. Melt formation resulting in the formation of micro-cavities on the surface of Ti alloy makes such structures interesting for the development of new implants [23].

Additionally, we presented a graph with *PO* and repetition rates that allows us to choose an appropriate texturing regime with preferred structures on Ti alloy depending on the application. Based on this graph, it is possible to move from pure trenches without melting to those formed by melting of the material depending on the applied repetition rate. In principle, these structures can be used for drug delivery systems on the surface of Ti alloy using ultrashort laser pulses in the kHz and MHz regimes.

In the future, our studies will focus on mechanical testing of textured Ti samples and the characterization of drug delivery properties of these structures.

## Figures and Tables

**Figure 1 materials-16-02498-f001:**
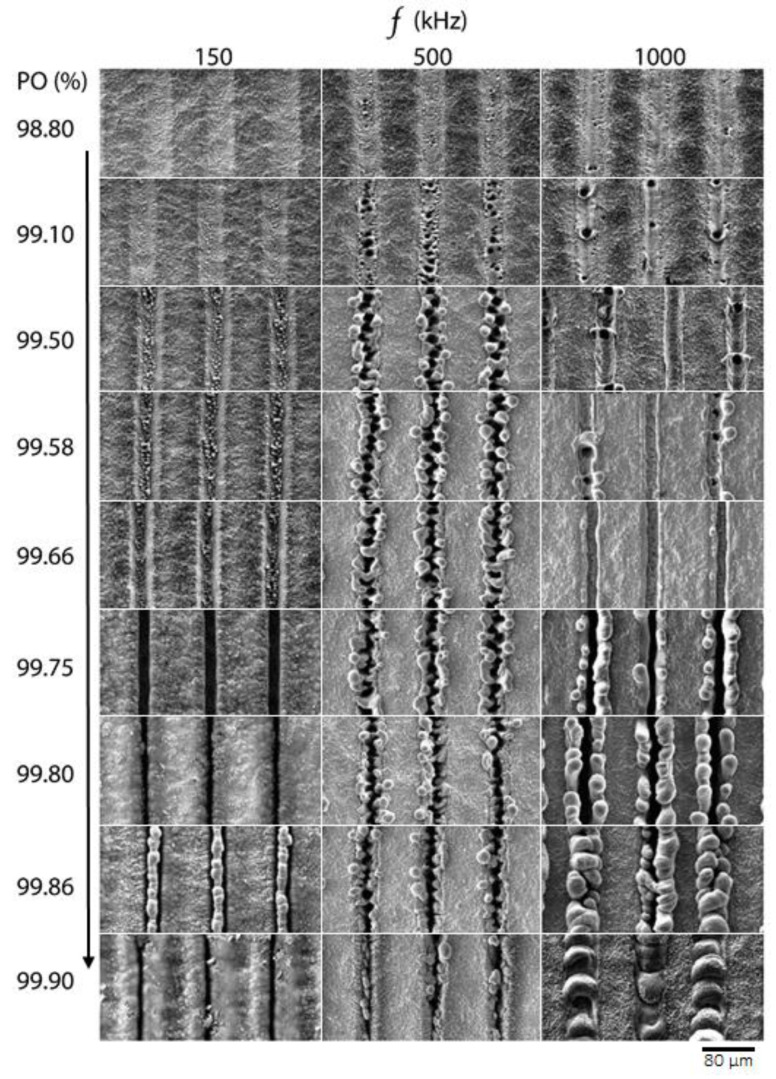
SEM images of structures on Ti alloy fabricated at 150 kHz, 500 kHz, and 1 MHz.

**Figure 2 materials-16-02498-f002:**
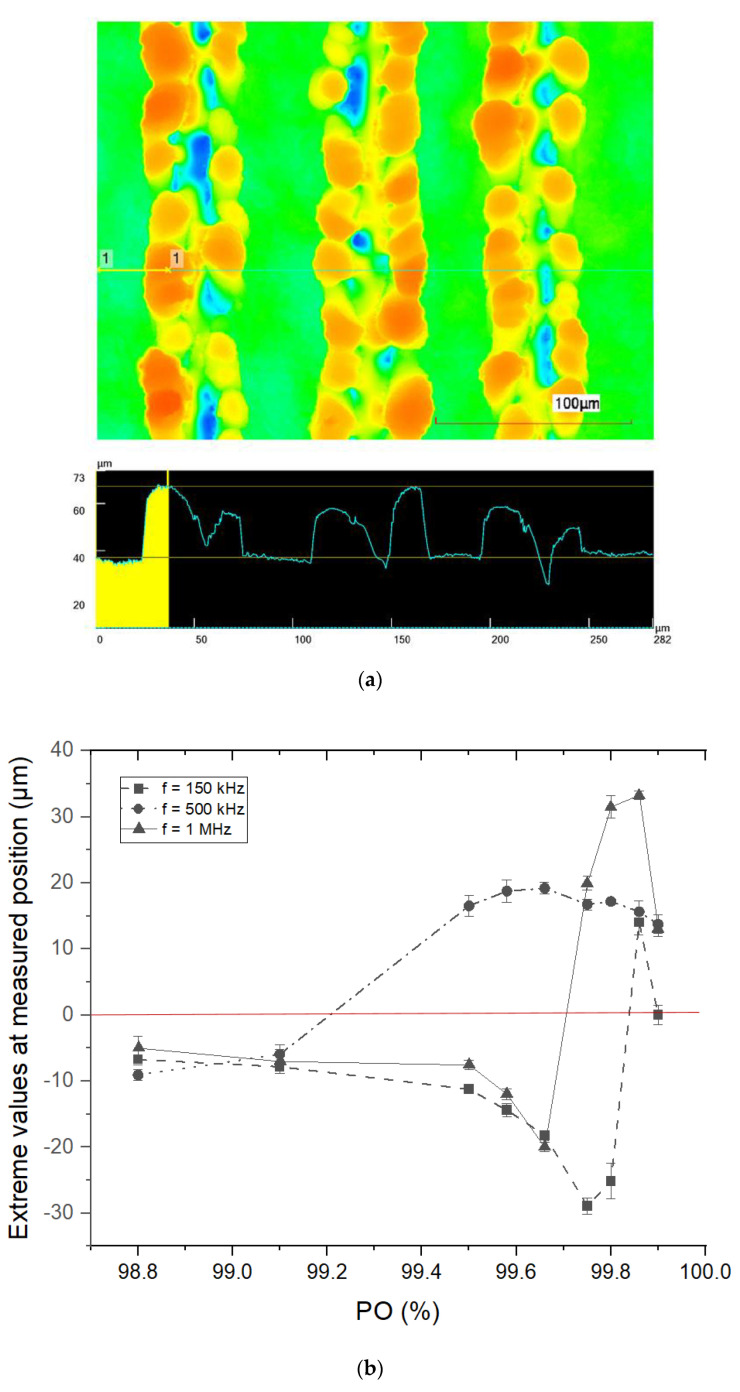
CLSM results of surface topography evolution during laser texturing: (**a**) Representative CLSM image of the surface profile at a measured position (black line); (**b**) mean extreme values at a measured position at *f* = 150 kHz, *f* = 500 kHz, and *f* = 1 MHz (measuring position and mean deviation from *N* = 5 measurements).

**Figure 3 materials-16-02498-f003:**
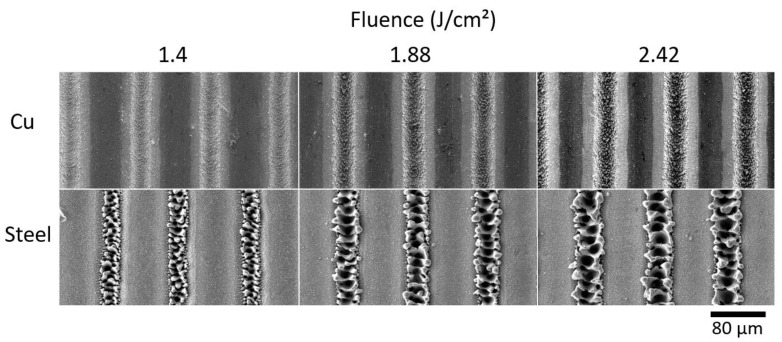
SEM images of microstructures on steel and copper fabricated with a repetition rate of 500 kHz at pulse overlap 99.93%.

**Table 1 materials-16-02498-t001:** Laser parameter variations used for Ti alloy texturing.

**Repetition rate (kHz)**	150/500/1000
**Pulse overlap (*PO*) (%)**	98.80 99.10 99.50 99.58 99.66 99.75 99.80 99.86 99.90
**Pulse energy (µJ)**	9.9
**Fluence (J/cm^2^)**	1.4

**Table 2 materials-16-02498-t002:** Laser parameter variations used for copper and steel texturing.

**Repetition rate (kHz)**	500
**Pulse overlap (*PO*) (%)**	99.33
**Pulse energy (µJ)**	9.9 13.3 17.1
**Fluence (J/cm^2^)**	1.4 1.88 2.42

**Table 3 materials-16-02498-t003:** Physical properties of textured metals.

	Ti Alloy	Cu	Steel
**Thermal conductivity** **(W/m/K)**	21.9 [52]	401 [52]	14.9 [52]
**E–ph coupling constant** **(10^17^ W/m³/K)**	18.5 [52]	0.48 [52]	31.7 [50]
**Ablation threshold** **(J/cm²)**	0.037 [53]	0.097 [53]	0.10–0.25 [54,55]

## Data Availability

Not applicable.

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
