# Peer review of "Heat Accumulation-Induced Surface Structures at High Degrees of Laser Pulse Overlap on Ti6Al4V Surfaces by Femtosecond Laser Texturing"

_materials, 2023, doi:10.3390/ma16062498_

Round 1

Reviewer 1 Report

1. The figure in Figure 1 and Figure 2 is separated from the figure title and needs to be reformatted.

Author Response

Dear reviewer,

thank you for taking time to review our article. We appreciate it very much.

As for your comment, thank you for pointing out an error. The text below the figures was apparently moved when we inserted our article into the journal's template. We have corrected it.

Yours sincerely,

All authors

Reviewer 2 Report

The author has nicely demonstrated the femtosecond laser texturing of Ti6AL4V surfaces induced by heat accumulation due to the effect of laser pulse overlap. However, the author is suggested to take minor revisions before being considered for publication.

1.     There are many grammatical as well as typo errors in the manuscript. For example, J/cm2 is should be J/cm2 in Table 2. The author is suggested to take careful revision of entire manuscript to avoid such error.

2.     The reviewer is interested to know on what basis the author is selected the laser process parameters for different materials as shown in Table 1 and Table 2.

3.     The author is strongly advised to mention the limitation of the present study in the conclusion section.

4.     The author is strongly advised to mention the future scope of the present research work in the conclusion section.

Author Response

Dear reviewer,

thank you for taking time to review our article. We appreciate it very much.

Below are our responses to your comments and suggestions.

  1. Thank you for pointing out the typo. We have also carefully revised the entire manuscript to avoid such errors.
  2. During our experiments, we found that melting of the Ti alloy occurs at POs above 98%, which is not the case below these POs. Therefore, we decided to conduct our experiments with the parameters listed in Table 1 and Table 2.
  3. We have considered your comments and included them in the conclusion.
  4. We have also included the future scope of our research in the conclusion.

Yours sincerely,

All authors

Reviewer 3 Report

The manuscript entitled “materials-2241349-Laser” dealing with laser processing has been reviewed. The paper has been nicely written but needs significant improvement. Please follow my comments.

1.     What is the main research question for this research work?

2.     Why 150 / 500 / 1000 were selected in Table 1?

3.     What is the future direction of this work?

4.     If possible please consider to change Figure 1 shows an overview of the surface structures generated via the laser para…

5.     Laser absorptivity is important which shows the quality of the parts and transition from keyhole to conduction mode. Please read and add the following ref in this area. “The effect of absorption ratio on meltpool features in laser-based powder bed fusion of IN718”.

6.     Please proofread the paper.

7.     Laser has many advantages over the conventional manufacturing method which can be highlighted in your paper. Please read the following manuscript and add it to the literature to show how the laser is comparable with conventional manufacturing.

·       Laser subtractive and laser powder bed fusion of metals: review of process and production features

Author Response

Dear reviewer,

thank you for taking time to review our article. We appreciate it very much.

Below are our responses to your comments and suggestions.

  1. The main research question of this work is to prove that it is possible to fabricate structures or micro-cavities on the surface of a Ti alloy that can be used in the implant industry. Implants that have micro-cavities can be filled with drugs prior to implantation. After implantation, the drugs act locally and prevent bacteria from adhering. In addition, these structures are fabricated with ultrashort laser pulses, resulting in a smaller heat-affected zone (HAZ) compared to nano- or picosecond lasers, which affects the mechanical properties of the material by changing the microstructure of the material and introducing residual stresses. We have revised our introduction to make it clearer.
  2. We used 150 kHz, 500 kHz, and 1000 kHz because our laser system operates at these frequencies. In addition, these repetition rates provide the time interval between successive laser pulses, which are 6.6 µs, 2 µs, and 1 µs, respectively.
  3. The future direction of this work is to perform mechanical tests on Ti specimens textured with these parameters. We have added this to the conclusions.
  4. Unfortunately, we cannot change the figure 1. Since it contains the images of 27 laser parameters, we found such a representation of the images easiest for the readers. Otherwise it would be too complicated to understand.
  5. Yes, that is true. We wrote that the formation of the resulted structures is a result of complex material interaction with incident laser beam that is dependent on electron-phonon (e-ph) coupling constant, thermal conductivity and other physical properties of the material, wavelength and polarization of the incident light, machining environment, and laser parameters, such as the laser fluence and degree of pulse overlap. We also added the laser absorptivity.
  6. We did.
  7. Yes, the laser systems have many advantages over the conventional manufacturing methods. We have mentioned some methods for surface modification of titanium in the introduction. These include sandblasting, chemical etching, plasma electrolytic oxidation, thermal oxidation and corundum grid blasting which severely reduce the fatigue performance of implants. We have also found that laser texturing reduces surface contamination while being environmentally friendly. Therefore, we believe that we have provided sufficient information on conventional manufacturing methods for titanium alloy surface modification processes.

Yours sincerely,

All authors

Reviewer 4 Report

1.Please explain the novelty of the current work

2. Typo in Table 3. Heading

3. What is the basis for choosing a particular texture?

4. Section 3.1, 3.2 -  heading to be revised. (Don’t simply mention the materials used)

5. Fig. 3 – micron marker to be added.

6. Authors can strengthen their discussion.

7. Introduction or discussion can use the following articles for strengthening the scientific rigour.https://doi.org/10.1007/s13369-022-07256-9, https://doi.org/10.1016/j.optlastec.2022.108210

8. Conclusions to be made pointwise.

Author Response

Dear reviewer,

thank you for taking time to review our article. We appreciate it very much.

Below are our responses to your comments and suggestions.

  1. The novelty of this work is to show that it is possible to fabricate structures or micro-cavities on the surface of a Ti alloy that can be used in the implant industry with ultrashort laser pulses on the surface of the Ti alloy. This results in a smaller heat-affected zone (HAZ) compared to nano- or picosecond lasers, which affects the mechanical properties of the material by altering the microstructure of the material and introducing residual stresses. Titanium implants with micro-cavities can be filled with drugs before implantation. After implantation, the drugs act locally and prevent the adhesion of bacteria. We have also addressed the cause of the formation of such structures. Although ultrafast laser texturing is thought to be melt-free, we show that this is not the case at high repetition rates. We have attributed such phenomena to local heat accumulation of the Ti alloy due to its low thermal conductivity.
  2. Corrected, thank you.
  3. Such a texture, which resembles micro-cavities on the surface of a Ti alloy, can be used in the implant industry. Implants with micro-cavities can be filled with drugs before implantation. After implantation, the drugs act locally and prevent bacteria from settling.
  4. Edited, thank you.
  5. Corrected, thank you for pointing out.
  6. We find it quite a detailed discussion. Which aspect should we strengthen? Could you please be more precise?
  7. Unfortunately, we cannot use the provided article to strengthen our work. The reason is that we use an ultrashort pulse laser system in our experiments, while a continuous wave (CW) laser is used in this article. Pulsed laser systems and CW laser systems are not comparable in any way. The mechanisms of interaction between laser and material are completely different.
  8. The representation of conclusions in the form of points is a matter of taste. So it would not change their meaning. Therefore, we have expanded our conclusions.

Yours sincerely,

All authors

Round 2

Reviewer 3 Report

The paper is in publishable format. 

Author Response

Dear reviewer,

thank you for your feedback.

Yours sincerely,

All authors

Reviewer 4 Report

All comments of my previous review are still not addressed. The authors seem to be arrogant in their way of response. Following are further clarifications on their manuscript. 

1. The authors does not even know the unit of El-ph coupling constant (Table 3). If we point this out we get an irritating response. 

2. Why is ablation threshold not given for steel?

3. Is it  2,42 J/cm² or 2.42 J/cm² in line 302 page 9?

4.  'laser-induced melt formation and the 277 Marangoni effect, in which the melt is driven from the center of the molten pool to the 278 edge, where it solidifies'  How is this confirmed in the present work?

5. Conclusions are too wordy. They need to be crisp and short. 

I would accept only if my previous comments and this comments are addressed and the authors respond in a humble and polite manner. Any stalwart should be able to accept one's mistake. The only he/she can be a stalwart by knowing more and more. 

Author Response

Dear reviewer,

thank you for taking time to review our article. First, let us apologize if we sounded impolite to you. This was not our intention and we are sorry if you understood it in such way. We are also sorry that our initial responses were not sufficient. Therefore, we expanded our corrections and hope that they fit your expectations.

Below you will find our responses to your comments and suggestions from your first review. Changes in the paper concerning your reviews are marked in yellow.

  1. Thank you for your question. This work demonstrates that femtosecond (fs) pulsed lasers can be utilized to create micro-cavities or structures of different geometries – even with undercuts due to melt reformation on the surface of a Ti alloy. Latter is a material typically applied in implant industry. This laser induced approach results in a high geometric flexibility combined with a smaller heat affected zone compared to pico- or nanosecond pulsed lasers that are industrially more relevant compared to fs-lasers. In this paper, we focused on the formation of different types of microstructures that can be fabricated with using fs-lasers. We found that at high repetition rates, local heat accumulation of the Ti alloy occurs due to its low thermal conductivity, even though ultrafast laser texturing is commonly believed to be melt-free. To prove our assumptions we also textured copper. Our experiments with copper have confirmed our hypothesis regarding heat accumulation on metals with low thermal conductivity, as we observed the absence of melt formation.

The minimization of thermal impact is a promising approach to ensure better mechanical properties for laser texturing applications, which is in focus of our current research. Additionally, by filling the micro-cavities in titanium implants with drugs prior to implantation, the drugs can act locally to prevent bacteria adhesion, minimizing off-target toxicity. Also. one of the other reviewers demanded a short outlook on these topics and we added it to our conclusion in a short way.

  1. We thank the reviewer for his reading and remark to this error. We have changed it manuscript.
  2. Thank you for your question. Concerning the described medical application (see response to remark 1.) literature describes micro-cavities with undercuts to be very suitable for a controlled drug release. Therefore, the focus in the given paper was to study fs-pulsed laser structuring processes to realize similar structure geometries, which is only possible via melt reforming during laser texturing.

In order to clarify this in the text the following additions were made:

Lines 66-67: “A good control of drug release properties was achievable via microstructures with under-cuts.”

Lines 88-89: “[…], enabling also the generation of microstructures with undercuts.”

  1. We thank the reviewer for his remark and changed the headings according to the following:

3.1: “Microstructure formation on titanium at high pulse repetition rates”

3.2: “Influence of substrate material on melt formation”.

  1. We have to apologize, that we missed this. We added the scale bar to the figure. Thank you for your remark.
  2. We thank you for this remark and assume it is in context with remark 4 from your second revision. So thank you again for clarifying your first remark. The observations concerning parameter depending structure formation we describe were done under pulsed laser texturing conditions that are not well studied so far. Yet, mechanisms that can be helpful to explain these observations can be derived from other experiments from the literature. Therefore, we are discussing our observations very closely to the literature. Within this discussion the Marangoni effect was described by Liu et. Al. (Ref. 49) to be responsible for melt cast out during micro-hole texturing with a static laser beam. We found this research relevant to ours and decided to mention it in the discussion to strengthen it and remind of the possible reasons for the formation of structures together with the heat accumulation effect. We are sorry, that this mentioning implied, that we are proposing a direct relation between our observations and the Marangoni effect. It was only meant as a possible mechanism for structure formation. We hope, that this explanation clarifies the topic and want to thank you again.
  3. We thank you for drawing our attention to these nicely written papers. They are presenting in a very nice manner the influence of laser parameters on melt formation and melt reforming during fiber laser cutting with a continuous wave (CW) laser. These CW lasers are very suitable for cutting applications, resulting in a pronounced melt formation due to high thermal impacts. Other than femtosecond-pulsed laser texturing the CW laser beam moves over the surface in a continuous way, without pulse-to-pulse distances and a defined pulse-pause-ratio, leading to local re-solidification effects, as it is desired for laser texturing. Therefore, we have to regret, but we don’t see these papers as suitable for adding to our discussion. We hope, that you accept such explanation.
  4. Thank you very much for your remark. We see it together with remark 5 from your second revision. We agree with you, that the conclusion should be made in short and summarized way. In order to make our conclusion more crisp we deleted the sentences in lines 319 to 326:

“It should also be noted that not every ultrashort pulse laser system can produce such tex-tures. The laser repetition rate must be sufficient to achieve heat accumulation on the sur-face of the textured metal. Another important aspect is the choice of a metal with low thermal conductivity, as is the case with Ti alloys and steel. Otherwise, heat accumulation will not occur and the desired structures cannot be created, in the examined parameter interval. Current developments of ultrashort laser pulse systems which operate at high repetition rates enable local melt formation on the surface of textured Ti alloy.”

To fill the resulting gap we added the short sentence in lines 326 to 328: “This effects have to be taken into account for specific laser system and material combinations, especially for novel developments in high repetition ultra short pulsed lasers.”

Yet, we politely mention that doing our conclusion in a pointwise way it could result in a loss of significant information to the readers that could only be sufficiently clarified in a fluent text. Additionally, other reviewers suggested to include the future scope of our work and its limitations to our conclusion, which could be better done in the given way. We have not found any indications to special requirements for MDPA “Materials” with respect to this topic. Therefore, we are directly addressing the editor and will be happy to adapt the conclusion to her requirements to be suitable for the readers of the journal. We hope that these changes and our approach fits to your wishes.

Second revision:

  1. Thank you for pointing out the error. We have corrected in the text.
  2. Thank you very much for your remark. We did not included it the first time because we saw it not necessary for our discussion. We added it now for better explanation, including literature references (see Figure 3).
  3. Thank you for reading our paper in such a detailed manner. We appreciate it at feel sorry for this error. It was corrected.
  4. Please see remark 6. from your first revision.
  5. Please see remark 8. from your first revision.

Yours sincerely,

All authors